# Electrical and Structural Characteristics of Excimer Laser-Crystallized Polycrystalline Si_1−x_Ge_x_ Thin-Film Transistors

**DOI:** 10.3390/ma12111739

**Published:** 2019-05-29

**Authors:** Kyungsoo Jang, Youngkuk Kim, Joonghyun Park, Junsin Yi

**Affiliations:** College of Information and Communication Engineering, Sungkyunkwan University, 2066 Seobu-ro, Jangan-gu, Suwon-si, Gyeonggi-do 16419, Korea; jks30716@skku.edu (K.J.); bri3tain@skku.edu (Y.K.)

**Keywords:** poly-Si_1−x_Ge_x_, excimer laser annealing, thin-film transistor, density of states

## Abstract

We investigated the characteristics of excimer laser-annealed polycrystalline silicon–germanium (poly-Si_1−x_Ge_x_) thin film and thin-film transistor (TFT). The Ge concentration was increased from 0% to 12.3% using a SiH_4_ and GeH_4_ gas mixture, and a Si_1−x_Ge_x_ thin film was crystallized using different excimer laser densities. We found that the optimum energy density to obtain maximum grain size depends on the Ge content in the poly-Si_1−x_Ge_x_ thin film; we also confirmed that the grain size of the poly-Si_1−x_Ge_x_ thin film is more sensitive to energy density than the poly-Si thin film. The maximum grain size of the poly-Si_1−x_Ge_x_ film was 387.3 nm for a Ge content of 5.1% at the energy density of 420 mJ/cm^2^. Poly-Si_1−x_Ge_x_ TFT with different Ge concentrations was fabricated, and their structural characteristics were analyzed using Raman spectroscopy and atomic force microscopy. The results showed that, as the Ge concentration increased, the electrical characteristics, such as on current and sub-threshold swing, were deteriorated. The electrical characteristics were simulated by varying the density of states in the poly-Si_1−x_Ge_x_. From this density of states (DOS), the defect state distribution connected with Ge concentration could be identified and used as the basic starting point for further analyses of the poly-Si_1−x_Ge_x_ TFTs.

## 1. Introduction

Polycrystalline silicon (poly-Si) thin-film transistors (TFTs) are widely used for the backplane of display devices, such as active matrix liquid crystal displays (AMLCD) or active matrix organic light-emitting diodes (AMOLEDs), because their field-effect mobility (µ_FE_) and electrical stability are superior to those of hydrogenated amorphous silicon (a-Si:H) TFTs [1,2]. However, the fabrication cost of poly-Si TFTs, which can be achieved via an additional annealing process, such as excimer laser annealing (ELA), solid-phase crystallization (SPC), and metal-induced crystallization (MIC), is higher than that for a-Si:H [3,4]. ELA is among the most popular methods used to crystallize a-Si:H to form poly-Si [5,6], and the process can be briefly explained as follows: high-energy pulsed laser beams are absorbed into the a-Si:H layers and produce localized heating, resulting in melting and recrystallization, which leads to the formation of poly-Si. Moreover, a high-energy laser beam ensures localized heating without a significant spreading of the temperature to other areas of the TFT or substrates. However, even the high mobility of the excimer laser-annealed poly-Si TFT is not sufficient for many circuit applications and, hence, further enhancement of the mobility is necessary for high-level system integration [7].

Polycrystalline silicon–germanium (poly-Si_1−x_Ge_x_) can be considered as a potential active channel layer for TFT applications because the poly-Si_1−x_Ge_x_ thin film has a narrow optical bandgap (0.8 eV) and higher carrier mobility compared to that of poly-Si. Normally, the small bandgap material has a high mobility. It was observed that the mobility of poly-Si with a small bandgap (1.1 eV) is higher than that of amorphous silicon (1.8 eV) [8]. Furthermore, amorphous zinc oxynitride (ZnON) with small bandgap (1.3 eV) has a high intrinsic mobility compared with high-bandgap materials such as ZnO (3.1 eV) [9]. At room temperature, the hole carrier mobility of Si is 475 cm^2^/Vs, while that of Ge is 1900 cm^2^/Vs [10]. However, the field-effect mobility of poly-Si and poly-Si_1−x_Ge_x_ TFT can be changed via the fabrication process. At room temperature, the hole carrier mobility of Si is 475 cm^2^/Vs, while that of Ge is 1900 cm^2^/Vs [8]. Therefore, a Ge-added Si thin film can be considered as a promising material for a high-performance device that can provide a very-high-definition display. Moreover, the poly-Si_1−x_Ge_x_ alloy can be easily achieved using a GeH_4_ and SiH_4_ gas mixture for chemical vapor deposition. However, the poly-Si_1−x_Ge_x_ TFTs are mostly fabricated via conventional SPC, whereas the use of ELA is rarely reported [11,12,13]. Furthermore, the characteristics of poly-Si_1−x_Ge_x_ with different Ge concentrations and properties, such as grain size and surface roughness, are not properly reported. Hence, there is no reported optimization of excimer laser density conditions and Ge concentrations to achieve the fabrication of high-performance poly-Si_1−x_Ge_x_ TFTs with desirable characteristics.

In this study, we investigated the characteristics of poly-Si_1−x_Ge_x_ thin films with different Ge concentrations and analyzed their electrical characteristics. It was reported that a Ge content of less than 15% has higher mobility than it does otherwise. Therefore, the study was conducted on less than 15% of Ge content (0–12.3%) [14]. The Ge concentration was varied using different GeH_4_/H_2_ gas flow ratios, and the a-Si_1−x_Ge_x_ was crystallized using an excimer laser. To optimize the performance of the TFT, the excimer laser energy density was varied, and the resulting structural characteristics, such as grain size, roughness, and crystallinity, as well as the electrical characteristics, of the poly-Si_1−x_Ge_x_ thin film were analyzed. Finally, samples of the poly-Si_1−x_Ge_x_ TFTs with different Ge concentrations were fabricated, and their electrical performance was evaluated. In addition, we analyzed the effect of the density of states (DOS) modeling on the electrical performance of the poly-Si_1−x_Ge_x_ TFT using a technical computer-aided design (TCAD) simulator. This study is expected to elucidate the poly-Si_1−x_Ge_x_ TFT fabrication process and optimize the electrical characteristics of the TFTs.

## 2. Device Fabrication

To fabricate top-gated poly-Si_1−x_Ge_x_ TFTs, a glass substrate was prepared. After the deposition of 300-nm-thick SiO_2_ buffer layers on the glass, the a-Si_1−x_Ge_x_:H film of 50 nm thickness was deposited via plasma-enhanced chemical vapor deposition (PECVD). The deposition temperature and working pressure were 200 °C and 40 Pa, respectively. For the deposition of a-Si_1−x_Ge_x_:H films, silane (SiH_4_), germane (GeH_4_), and hydrogen (H_2_) were used. The SiH_4_/H_2_ gas flow ratio was fixed at 0.75, and the GeH_4_/H_2_ gas flow ratio was varied from 0 to 0.04. The poly-Si_1−x_Ge_x_ films were crystallized using pulsed XeCl (λ = 308 nm) excimer laser irradiation (20 shots). To avoid hydrogen atom ablation due to the high-energy irradiation, the energy density of the excimer laser was gradually increased from low to high energy density; the laser energy density was varied from 360 mJ/cm^2^ to 450 mJ/cm^2^ with a step of 30 mJ/cm^2^ for fabricating the samples. The Ge concentration was mainly affected by the GeH_4_ flow rate; thus, the Ge content (x) within the film was raised from 0 to 12.3%. The poly-Si_1−x_Ge_x_ thin film was observed using scanning electron microscopy (JEOL, JSM-6390A, Tokyo, Japan) Raman spectroscopy (Bruker, FRA 160/S, Billerica, MA, USA), and atomic force microscopy (SII, SPA-300HV, Chiba, Japan) to analyze its structural characteristics. After the formation of the poly-Si_1−x_Ge_x_ thin film, a 200-nm-thick SiO_2_ film was deposited as a gate dielectric via PECVD at 200 °C on the poly-Si_1−x_Ge_x_. Aluminum was deposited as a gate electrode via thermal evaporation, and it was patterned as a self-aligned structure. Then, the boron ion shower doping technique was applied to fabricate the p-type transistor, and the dopant activation process via excimer laser irradiation was performed at room temperature. The energy density for the activation process was similar to or slightly smaller than the crystallization process. The channel width and length of the fabricated TFTs were 180 and 50 µm, respectively, and the drain to source voltage (V_DS_) was maintained at −0.1 V. The electrical characteristics of the poly-Si_1−x_Ge_x_ TFTs were examined using a semiconductor parameter analyzer. The schematic structure of the fabricated TFT is shown in Figure 1.

## 3. Results and Discussion

For the crystallization of a-Si:H using ELA, three regimes were suggested in Reference [15]. The first regime involves low energy density, where the excimer laser can partially melt the surface of the a-Si. Because the energy density is not sufficient to crystallize the entire a-Si thickness, the grain size of the poly-Si is generally smaller than the thickness of the a-Si. The second regime uses a higher energy density, and a nearly complete regime occurs; there is a significant lateral growth of the a-Si. In this regime, the poly-Si is almost independent of the energy density and can be related to the homogeneous nucleation from a temperature gradient; the grain size is bigger than the thickness of the a-Si. The third regime, which is a complete melting regime, involves much higher energy density, and deep super-cooling is achieved followed by the nucleation and growth of solids. A fine-grained poly-Si and sometimes amorphized poly-Si due to the hydrogen explosion is obtained. The specific energy density for the three regimes is changed by a-Si or a precursor condition. In this study, we experimented with various excimer energy densities to optimize the energy conditions.

Figure 2 shows the images of the poly-Si_1−x_Ge_x_ grains, the average grain size for different Ge concentrations, and the energy density of the excimer laser. Before the SEM measurement, the widely used secco etching (a mixed solution of K_2_Cr_2_O_7_ and HF in water) was performed to observe the poly-Si_1−x_Ge_x_ crystalline grains and the amorphous grain boundaries. The grain size of the poly-Si_1−x_Ge_x_ was varied by different energy densities and Ge concentrations; however, the grain size was slightly changed in the same conditions. To compare the average grain size, the number of poly-Si_1−x_Ge_x_ grains per squared area was estimated. A complete grain in the squared area was assumed as one grain, and the partial grains were assumed as 0.5 grains. In the case of 0% Ge, which is poly-Si, the grain size was observed to increase with increasing laser energy density, and the largest grain size (about 363 nm) was obtained at a laser energy density of 450 mJ/cm^2^. Similarly, in the case of 5.1% and 7.1% Ge, the grain sizes were initially observed to increase with increasing laser energy density, and, at the energy density of 420 mJ/cm^2^, the largest grain sizes of 375 nm and 333 nm were obtained, respectively. However, at energy densities higher than 420 mJ/cm^2^, the grain sizes were observed to be smaller than 100 nm. This was also observed in the case of 10.3% and 12.3% Ge, for which the largest grain sizes of 301 nm and 324 nm were obtained at the energy density of 390 mJ/cm^2^; at higher energy densities, the grain sizes decreased rapidly with increasing energy density. Thus, it was observed that, when the Ge concentration was increased, the maximum grain size was obtained at a specific energy density, beyond which the grain size was rapidly reduced. This is generally associated with an increase in the grain size as the Ge volume increases and the crystalline volume fraction approaches 1 at lower laser energy densities [16]. For the poly-Si_1−x_Ge_x_ thin films, the images of the average grain size and the graphs show that the grain size is more sensitive than for the poly-Si. At a suitable energy density, the highest grain size is achieved, below or beyond which the grain size decreases. Thus, it was observed that the optimum energy density for super lateral growth is very closely related to the Ge content. Figure 3 shows the XPS Ge 3*d* peak, and Ge and Si components are inserted. Figure 3 shows the real concentration of the Ge content.

The growth of the grains at different Ge concentrations in the Si_1−x_Ge_x_ film can be explained by the difference in the thermal conductivity, which is very important in the laser crystallization process because the thermal gradient can cause lateral grain growth. The thermal conductivity of Ge (13 W/mK) is higher than that of Si (2.7 W/mK); thus, the grain nucleation velocity of Ge is higher than that of Si [17]. When the Ge concentration in the poly-Si_1−x_Ge_x_ is increased, the temperature gradient in the poly-Si_1−x_Ge_x_ also increases and the grain nucleation velocity is higher than that of the poly-Si, resulting in a smaller grain size.

Figure 4 shows the full width at half maximum (FWHM) extracted from the Raman spectra of poly-Si_1−x_Ge_x_ films formed with different laser energy densities and Ge contents. The reduced peak width of FWHM indicates a better crystalline lattice. As the Ge content was increased from 0% to 12.3%, the laser energy density value with the lowest FWHM decreased from 450 mJ/cm^2^ to 390 mJ/cm^2^. This shows a good correlation between the SEM and Raman poly-Si_1−x_Ge_x_ film characterization. In addition, the Raman peak position was observed to vary with the Ge content. In this study, it was located at 518.3 cm^−1^ for the poly-Si thin-film, and, for the Si_0.877_Ge_0.123_ film, which contains Ge 12.3%, it was located at 511.1 cm^−1^. As the Ge concentration increased, the crystalline peak position shifted to righthand side. It was observed that the variations in the laser energy density had little effect in this study.

Figure 5 shows the surface morphology and root-mean-square (RMS) roughness of the poly-Si_1−x_Ge_x_ (0 ≤ x ≤ 0.123) films. The surface roughness of the poly-Si_1−x_Ge_x_ film was observed to increase with increasing Ge content. While the poly-Si (0% Ge) film exhibited the smoothest morphology with a roughness of 1.61 nm, the surface roughness of the poly-Si_1−x_Ge_x_ was observed to increase from 1.61 nm to 10.12 nm as the Ge content was increased from 0 to 12.3%. It is known that surface roughness has an important influence on the mobility of the TFT devices [18]. Therefore, in this study, Si_1−x_Ge_x_ thin films with a Ge content of up to 12.3% are expected to have low mobility when fabricating TFT devices due to the increased roughness. As shown in Figure 1, the thin film having 5.1% Ge content had the largest grain size of about 375 nm, which is expected to exhibit high mobility. However, since the surface roughness is more than four times that of the poly-Si thin film, it is expected that the mobility of the TFT device will be influenced. Thus, the electrical performance of the TFT may be degraded because of the poor interface between the poly-Si_1−x_Ge_x_ and the gate dielectric.

Figure 6 shows the transfer characteristics of the poly-Si_1−x_Ge_x_ TFTs with different Ge contents and laser energy densities for which the largest grain size was obtained (0%–450 mJ/cm^2^, 5.1%–420 mJ/cm^2^, 7.1%–420 mJ/cm^2^, 10.3%–390 mJ/cm^2^, and 12.3%–390 mJ/cm^2^). The width and length of the poly-Si_1−x_Ge_x_ TFTs were 180 and 50 µm, respectively (width/length = 3.6). Considerable changes in the electrical characteristics were observed at different Ge concentrations. The parameters with most noticeable differences were the on-and-off current ratio (I_ON_/I_OFF_), the threshold voltage (V_TH_), and the sub-threshold swing (SS). The I_ON_/I_OFF_ of the TFT for Ge 0%–450 mJ/cm^2^ was 7.56 × 10^6^, while that of the TFT with Ge 12.3%–390 mJ/cm^2^ was 1.28 × 10^5^, exhibiting a 59-fold decrease. The main reason for the variation in the on current is the grain size of the poly-Si_1−x_Ge_x_ TFTs; the Ge concentration induces poor crystallization quality and results in smaller grain sizes. Similarly, the V_TH_ of the TFT with Ge 0%–450 mJ/cm^2^ was −2.65 V, which decreased to −2.36 V for the TFT with Ge 5.1%–420 mJ/cm^2^. Then, it rapidly increased to −10.65 V for the TFT with Ge 12.3%–390 mJ/cm^2^. The SS was 0.75 V/decade for the TFT with Ge 0%–450 mJ/cm^2^, which slightly decreased to 0.68 V/decade for the TFT with Ge 5.1%–420 mJ/cm^2^, and then rapidly increased to 2.72 V/decade for the TFT with 12.3%–390 mJ/cm^2^. Furthermore, the electrical characteristics were thought to be influenced by the grain size, FWHM, and roughness; the electrical properties deteriorated due to the variations in the structural characteristics of the thin films as the Ge% increased. However, in the case of the poly-Si_0.949_Ge_0.051_ TFT, the V_TH_ and SS were slightly improved compared to those of the poly-Si TFT. To improve device performance in poly-Si TFT, many studies evaluated the quality of poly-Si. For poly-Si TFTs, the electrical characteristics, such as field-effect mobility or sub-threshold swing, are determined by the presence of grain boundaries in the channel region. The mixed-phase grain boundaries impede the electron flow electrically and morphologically; thus, it is one of the key parameters in the poly-Si TFT. However, in the poly-Si_1−x_Ge_x_ TFTs, the Ge distribution and concentration strongly influence the electrical performance. The Ge concentration influences the crystal structure, the grain size, and the roughness of the poly-Si_1−x_Ge_x_. The sub-threshold swing is closely connected with interface characteristics. Therefore, the quality of the poly-Si_1−x_Ge_x_/SiO_2_ interface is poorer than that of poly-Si/SiO_2_.

The field-effect mobility in the linear region was extracted from the maximum transconductance *g_m_*_,*MAX*_ as
$$g_{m,\;MAX}=\frac{\partial I_{DS}}{\partial V_{GS}},\;\mathrm{and}\;\mu FE\;=\frac{g_{m,\;MAX}L}{WC_{OX}V_{DS}},$$
where the field-effect mobility is proportional to *g*_m,*MAX*_ measured at a *V*_DS_ of 0.1 V. In this work, the field-effect mobility of the poly-Si TFT was 113.8 cm^2^/Vs, and the field-effect mobility obtained by employing Ge 5.1% was 66.8 cm^2^/Vs.

Thus, considering the structural characteristics of the thin film and the electrical characteristics of the TFT device, less than 5.1% Ge content should be used, and the TFT should be fabricated and compared with the poly-Si TFT.

Figure 7 shows the density of states (DOS) of the poly-Si_1−x_Ge_x_ TFTs extracted using the TCAD simulator. The DOS can be defined as a function of energy, as follows [19]:(1)$$g\left(E\right)=\mathrm{NTAexp}\left[\frac{E-E_{C}}{\mathrm{WTA}}\right]+\mathrm{NTDexp}\left[\frac{E_{V}-E}{\mathrm{WTD}}\right]+\mathrm{NGAexp}\left[-{\left(\frac{\mathrm{EGA}-E}{\mathrm{WGA}}\right)}^{2}\right]+\mathrm{NGDexp}\left[-{\left(\frac{E-\mathrm{EGD}}{\mathrm{WGD}}\right)}^{2}\right],$$
where E is the trap energy, E_C_ is the conduction band energy, E_V_ is the valence band energy, and the subscripts T, G, A, and D stand for tail, Gaussian (deep level), acceptor, and donor states respectively. The I–V characteristics of the TFT in Figure 4 show that the variations in the properties of the on current and SS depend on the Ge content and laser energy density. Therefore, this simulation focused on the DOS analysis related to the on current and SS. As the Ge contents increased, the number of the donor-like tail states increased. Moreover, for the tail states associated with the on current, the DOS value increased as the Ge concentration increased; for the case of Ge 0%–450 mJ/cm^2^, the density of the donor-like tail-state defects (NTD) at valence band maximum was 6 × 10^20^ cm^−3^∙eV^−1^; for the case of Ge 12.3%–390 mJ/cm^2^, the value of NTD was 9 × 10^21^ cm^−3^∙eV^−1^, which is 15 times larger; for the deep states associated with the SS, the donor-like deep-state defects (NGD) value increased as the Ge concentration increased as in the case of the tail states. Similarly, for the case of Ge 7.1%–420 mJ/cm^2^, the value of NTD was 6.5 × 10^21^ cm^−3^∙eV^−1^ and that of NGD was 2 × 10^18^ cm^−3^∙eV^−1^; for the case of Ge 10.3%–390 mJ/cm^2^, the value of NTD was 8 × 10^21^ cm^−3^∙eV^−1^ and that of NGD was 4.5 × 10^18^ cm^−3^∙eV^−1^. For the case of Ge 12.3%–390 mJ/cm^2^, the value of DOS at 0.3 eV was 7 × 10^18^ cm^−3^∙eV^−1^, which is seven times larger; at 0.3 eV, the values of the DOS were the same for Ge 5.1%–420 mJ/cm^2^ and Ge 0%–450 mJ/cm^2^. Furthermore, as reported in Reference [11], there is not much difference in the deep-level trap density between 4.5% poly-Si_1−x_Ge_x_ TFT, Ge 5.1% poly-Si_1−x_Ge_x_ TFT, and poly-Si TFT. Furthermore, it was observed that the defect states in the poly-Si_1−x_Ge_x_ TFT were higher at the overall energy band compared to that in the poly-Si TFT [20]. There are many reasons for this; primarily, as the Ge concentration is increased, the grain size is decreased and, thus, the grain boundary effect related to the defect states in the energy band is increased. Therefore, a higher number of donor-like tail states in the poly-Si_1−x_Ge_x_ thin-films is believed to result in higher microscopic strain. The heterostructure formed by the Si and Ge atoms could cause strain because the two have different properties, such as lattice constant, effective mass, and bond strength [21].

## 4. Conclusions

Poly-Si_1−x_Ge_x_ thin films reportedly have high field-effect mobility and, hence, are expected to produce high-performance TFTs for next-generation display devices. Poly-Si_1−x_Ge_x_ thin films crystallized via pulsed ELA are rarely reported in previous literature. In this study, we investigated the characteristics of excimer laser-annealed poly-Si_1−x_Ge_x_ thin films and their TFTs. To evaluate the influence of Ge concentration, it was varied from 0 to 12.3%. We experimented with various excimer energy densities to produce poly-Si_1−x_Ge_x_ with bigger grain sizes. The specific energy density to obtain super lateral growth conditions was observed to depend on the Ge concentration. This was attributed to the difference in the thermal conductivities of Si and Ge; the thermal conductivity of Ge is higher than that of Si; thus, the grain nucleation velocity in the poly-Si_1−x_Ge_x_ is higher than that in the poly-Si thin film. The Raman spectroscopic measurements showed that, as the Ge concentration increased, the laser energy density with the lowest FWHM decreased. Furthermore, it was observed that, as the Ge concentration increased, the surface roughness increased. Poly-Si_1−x_Ge_x_ TFTs with different Ge concentrations were fabricated. As the Ge concentrations increased, the electrical characteristics of the poly-Si_1−x_Ge_x_ TFTs, such as on current and sub-threshold swing, were degraded. The main reason for that was the grain size of the poly-Si_1−x_Ge_x_ TFTs resulting from the Ge concentration deceasing, which induced poor crystallization quality and smaller grain size. The transfer characteristics of poly-Si_1−x_Ge_x_ TFT were fitted in a TCAD simulation using the density of states distribution. The on current was affected by the density of the donor-like tail-state defects (NTD) near the valance band, while the sub-threshold swing and threshold voltage were affected by the donor-like deep-state defects (NGD). The results also showed that the defect states of the poly-Si_1−x_Ge_x_ TFTs were higher at the overall energy band compared to those of the poly-Si TFT. This may be caused by insufficient hydrogenation of the Ge dangling bond at the poly-Si_1−x_Ge_x_ and poly-Si_1−x_Ge_x_/SiO_2_ interface. Therefore, hydrogenation of the Ge dangling bond is required to improve both the field-effect mobility and sub-threshold swing for high-performance poly-Si_1−x_Ge_x_ TFTs.

## Figures and Tables

**Figure 1 materials-12-01739-f001:**
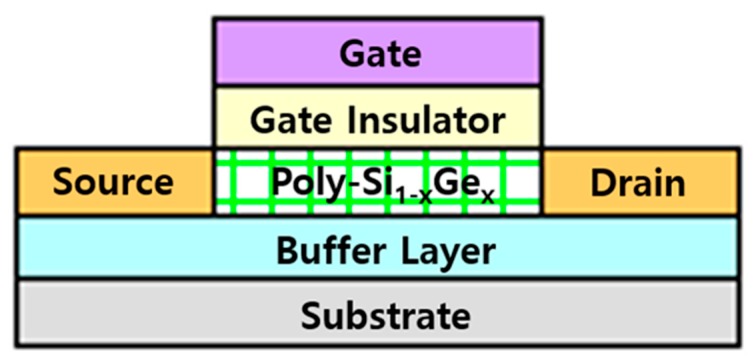
The schematic structure of the thin-film transistor (TFT) with polycrystalline silicon–germanium (poly-Si_1−x_Ge_x_) as an active layer.

**Figure 2 materials-12-01739-f002:**
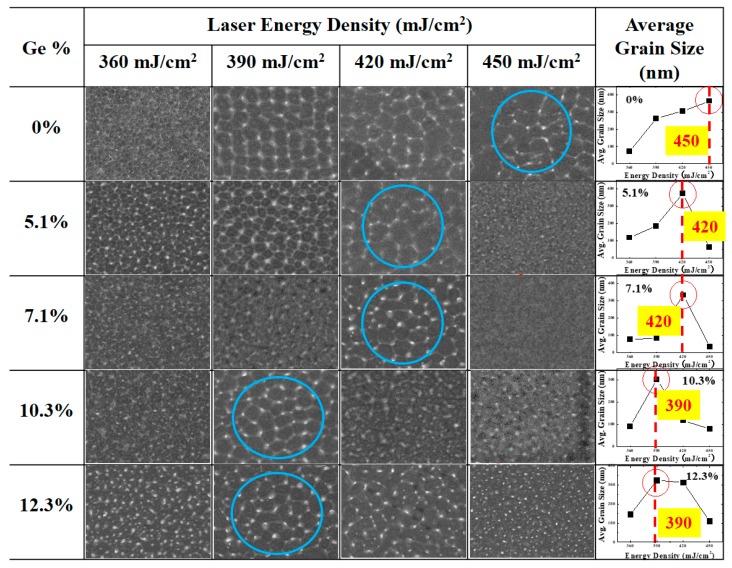
Grain images of poly-Si_1−x_Ge_x_ films as a function of the laser energy density at different Ge concentrations. The largest grain sizes are shown in circles.

**Figure 3 materials-12-01739-f003:**
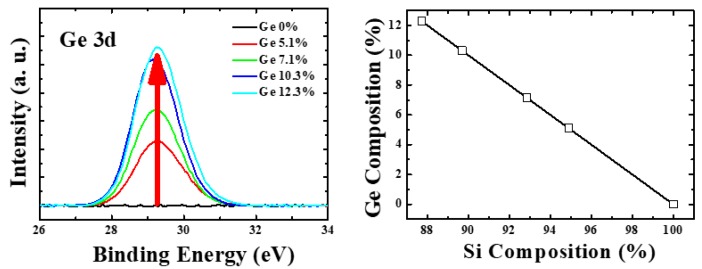
(**left**) X-ray photoelectron spectrum (XPS) of Ge 3*d* peak; (**right**) Ge and Si component ratio of the poly-Si_1−x_Ge_x_ thin film.

**Figure 4 materials-12-01739-f004:**
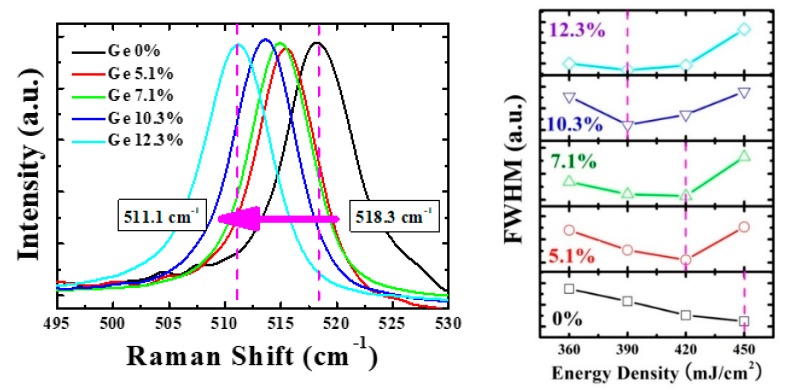
(**left**) Raman spectra of poly-Si_1−x_Ge_x_ thin films (**right**); full width at half maximum (FWHM) characteristics extracted from the Raman spectra of poly-Si_1−x_Ge_x_ films formed with different laser energy densities and Ge contents.

**Figure 5 materials-12-01739-f005:**
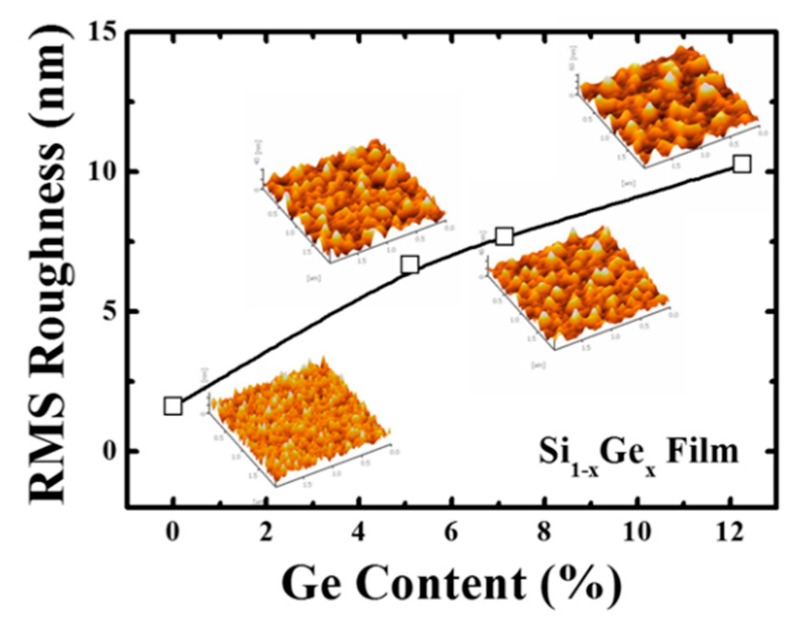
Surface morphology and root-mean-square (RMS) roughness of the poly-Si_1−x_Ge_x_ films as a function of Ge content. The surface roughness increases with increasing Ge content.

**Figure 6 materials-12-01739-f006:**
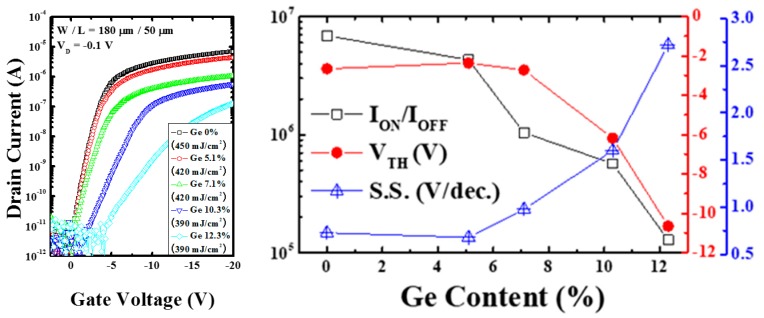
(**left**) Transfer characteristics; (**right**) electrical performance of poly-Si_1−x_Ge_x_ TFTs as a function of the laser energy density and Ge concentration.

**Figure 7 materials-12-01739-f007:**
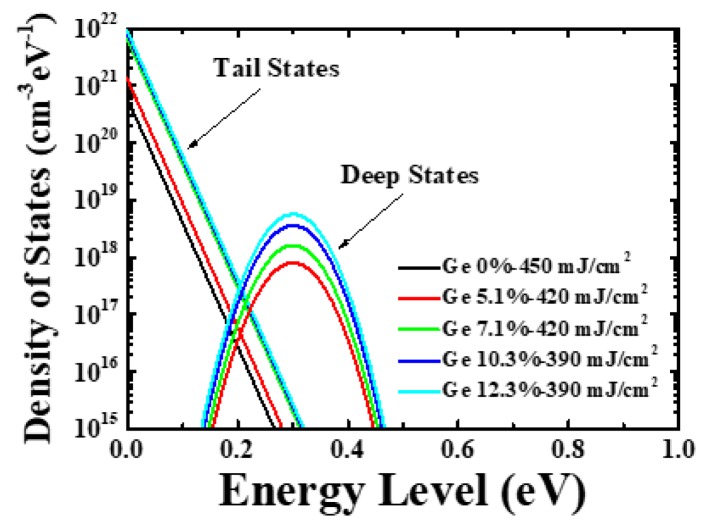
Distribution of density of states (DOS) as a function of the laser energy density and Ge concentration.

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
