# Peer review of "Electrical and Structural Characteristics of Excimer Laser-Crystallized Polycrystalline Si1−xGex Thin-Film Transistors"

_materials, 2019, doi:10.3390/ma12111739_

Round 1
Reviewer 1 Report
The materials aspects are of interest but the device characterisation is incomplete and therefore inconclusive. I suggest a more detailed study of the devices is required, therefore. Furthermore, the motivation for the adoption of poly-SiGe TFTs for displays etc needs to be made more convincingly.
42 Please state why a smaller bandgap is an advantage compared to poly-Si
43 The hole mobilities of poly-Si and poly-SiGe should be compared, not single Xtal.
158 The electrical performance will be degraded – not ‘may be’. Note that mobility needs to be extracted and reported. Also, what about surface (interface) states?
178 ‘thought to be influenced by the grain size’. This statement is too vague. Please consider in more depth the reasons for the relatively poor electrical behaviour. What about the nature of the gate oxide and its interface with the SiGe active layer? A more detailed study is required to investigate this aspect which is usually the biggest issue with devices of this type. C-V plots etc can be performed. Devices of different SiGe thickness could be used to identify the dominant source of SS degradation, namely the bulk defect distributions as modelled in the paper, or interface states.
Furthermore, carrier mobility should be extracted using the standard MOSFET models as is customary with such TFT devices. A comparison should be made to state of the art poly-Si devices.
189 Explain how the DOS parameter values were ‘extracted’. Presumably equation (1) was fitted in some way; details and a diagram is required.
196 Gaussian
221 The conclusions are vague with regard to the device performance. A more detailed study is required to elucidate the influence of Ge concentration on energy state distributions.
Author Response
The materials aspects are of interest but the device characterisation is incomplete and therefore inconclusive. I suggest a more detailed study of the devices is required, therefore. Furthermore, the motivation for the adoption of poly-SiGe TFTs for displays etc needs to be made more convincingly.
42 Please state why a smaller bandgap is an advantage compared to poly-Si
Thanks for kind suggestion. The manuscript was revised.
Normally, the small bandgap material has a high mobility. It is observed that mobility of poly-Si with a small bandgap (1.1eV) is higher than that of amorphous silicon (1.8 eV) [8]. Furthermore, amorphous zinc oxynitride (ZnON) with small bandgap (1.3 eV) has a high intrinsic mobility compared with high bandgap such as ZnO (3.1eV) [9].
43 The hole mobilities of poly-Si and poly-SiGe should be compared, not single Xtal.
Thanks for kind suggestion. The manuscript was revised.
At room temperature, the hole carrier mobility of Si is 475 cm2/Vs, while that of Ge is 1900 cm2/Vs [10]. But, the field effect mobility of poly-Si and poly-SiGe TFT can be changed by the fabricated process.
The electrical performance will be degraded – not ‘may be’. Note that mobility needs to be extracted and reported. Also, what about surface (interface) states?
- Thanks for kind suggestion. The electrical performance was shown in Figure 5(b).
178 ‘thought to be influenced by the grain size’. This statement is too vague. Please consider in more depth the reasons for the relatively poor electrical behaviour. What about the nature of the gate oxide and its interface with the SiGe active layer? A more detailed study is required to investigate this aspect which is usually the biggest issue with devices of this type. C-V plots etc can be performed. Devices of different SiGe thickness could be used to identify the dominant source of SS degradation, namely the bulk defect distributions as modelled in the paper, or interface states.
Thanks for kind suggestion. . The manuscript was revised.
To improve device performance in poly-Si TFT, a lot of researches were studied to the quality of the poly-Si. For poly-Si TFTs, the electrical characteristics, such as field effect mobility, or subthreshold swing, are determined by the presence of grain boundaries in the channel region. The mixed phase grain boundaries impede the electron flow electrically and morphologically, so that the it was one of the key parameter in poly-Si TFT. But, in the poly-Si1-xGex TFTs, Ge distribution and concentration strongly influence the electrical performances. Morever, it was reported that the phonon band structure so that it was revealed the Si–Si peak to appear at the right position whereas the Ge–Ge mode would be shifted to overestimate the Ge content. The subthreshold swing was closely connected with interface characteristics. So that the quality of the poly-Si1-xGex/SiO2 interface was presumed bad than poly-Si./SiO2.
- Furthermore, carrier mobility should be extracted using the standard MOSFET models as is customary with such TFT devices. A comparison should be made to state of the art poly-Si devices.
Thanks for kind suggestion. In the TFT industry, the field effect mobility was extracted by below equation. The field effect mobility in the linear region was extracted from the maximum transconductance gm,MAX as,
, and , (1)
where the field effect mobility is proportional to gm,MAX measured at a VDS of 1 V if L, W, VDS, and the channel oxide capacitance COX are fixed. In this work, the field effect mobility of the poly-Si TFT was 113.81 cm2/Vs and the field effect mobility obtained by employing Ge 5.1% is 66.77 cm2/Vs.
- 189 Explain how the DOS parameter values were ‘extracted’. Presumably equation (1) was fitted in some way; details and a diagram is required.
Thanks for kind suggestion. The equation was widely used for poly-Si TFT modeling. The detailed diagram was shown in silvaco inc. homepage
- 196 Gaussian
Thanks for kind suggestion. The manuscript was changed.
W stand for the width, T, G, A, D stand for band tail, gaussian deep, acceptor, and donor states respectively.
221 The conclusions are vague with regard to the device performance. A more detailed study is required to elucidate the influence of Ge concentration on energy state distributions.
Overall this works shows excellent progress and promise for improved understanding of passivation layer evaluation.
Thanks for kind suggestion. The manuscript was changed.
Poly-Si1-xGex TFT with different Ge concentrations was fabricated. As the Ge concentrations were increase, the electrical characteristics of the Poly-Si1-xGex TFT, such as on current and sub-threshold swing, were degraded. The main reason for that was the grain size of the poly-Si1-xGex TFTs resulting from the Ge concentration decease, and it induced poor crystallization quality and smaller grain size. The transfer characteristics of LTPS TFT were fitted in TCAD simulation using the density of states distribution. The on current and field effect mobility was affected by the density of the donor like tail state defects (NTD) near the valance band, while the sub-threshold swing and threshold voltage was affected by the donor like deep state defects (NGD). The results also showed that the defect states of the poly-Si1-xGex TFT were higher at the overall energy band compared to those of the poly-Si TFT. This may be caused by insufficient hydrogenation of the Ge dangling bond at the Poly-Si1-xGex and Poly-Si1-xGex/SiO2 interface. Therefore, it will be required hydrogenation of the Ge dangling bond to improve both the field-effect mobility and s-values for high performance Poly-Si1-xGex TFT.

Reviewer 2 Report
1.The Ge content of thin film were 0%、5.1%、7.1%、10.3%、12.3%. Why did authors decide to use Ge content from 0% to 12.3% instead of others? Authors should explain more explicitly in the introduction part.
2. In line 65, Please show the schematic structure of the final assembled device.
3. In line 103, Figure 1 shows the images of the poly-Si1-xGex grains, the average grain size for different Ge concentrations, but we don’t know the real concentrations in thin film. Please show XPS characterization of Si1-xGex thin film.
.
4. In line 141, the authors state that” the crystalline peak of Si-Si was observed decrease, but amorphous peak was decrease.” how do we know amorphous peak was decrease? Also, please show Raman spectra of Si1-xGex thin film.
5. It is recommended that authors make the graph of IdVd、SS、Vth. Maybe, it will be helpful for readers to analysis the electrical properties.
6. From Ref [11],we know the poly-Si- 0.91Ge- 0.09-TFT exhibited a high-hole mobility of 112 cm 2 /V-s. In line 153,Please list the mobility of.Si1-xGex thin film. It will help other research groups to reproduce more precisely the result of your work.
7. In line 175,It is known that threshold voltage of p-type transistor is negative. Therefore, it should be -10.65V for the TFT with Ge 12.3%-390 mJ/cm2.
8.In line 204,What are the value of DOS for the case of Ge 7.1% and 10.3%?
9.In line 211, the authors state that” as the Ge concentration is increased, the grain size is decreased; and thus, the grain boundary effect related to the defect states in the energy band is increased. ” What is the distribution of DOS as a function of Ge 10.3%-390 mJ/cm2 ? Is it really higher than the case of Ge 7.1%? Please show it in Figure 5.
Author Response
1.The Ge content of thin film were 0%、5.1%、7.1%、10.3%、12.3%. Why did authors decide to use Ge content from 0% to 12.3% instead of others? Authors should explain more explicitly in the introduction part.
However some major points must be taken into consideration:
Thank you very much for your valuable comment. Following your argument, we would like to explain as follows:
It was reported that the Ge content of less than 15% has a higher mobility than it does otherwise. Therefore, the study conducted on less than 15% of Ge content(0~12.3%) [14].
This is inserted in the introduction part.
2. In line 65, Please show the schematic structure of the final assembled device.
- Thanks for kind suggestion. The schematic structure of the final device is shown in device fabrication part as follows. (Figure 1)
Figure 1. The schematic structure of the device with poly-Si1-xGex as an active layer.
3. In line 103, Figure 1 shows the images of the poly-Si1-xGex grains, the average grain size for different Ge concentrations, but we don’t know the real concentrations in thin film. Please show XPS characterization of Si1-xGex thin film.
- Thanks for kind suggestion. The Figure 3 was added.
The Figure 3 shows XPS Ge3d peak and Ge and Si components are inserted. The Figure 3 shows real concentration of the Ge content.
4. In line 141, the authors state that” the crystalline peak of Si-Si was observed decrease, but amorphous peak was decrease.” how do we know amorphous peak was decrease? Also, please show Raman spectra of Si1-xGex thin film.
- Thanks for kind suggestion. Description of additional references was summarized. And, manuscript was changed.
As the Ge concentration was increased, the crystalline peak position was shifted to right side.
5. It is recommended that authors make the graph of IdVd、SS、Vth. Maybe, it will be helpful for readers to analysis the electrical properties.
- Thanks for kind suggestion. As your comment, electrical characteristics are shown in figure 6 (right).
6. From Ref [11],we know the poly-Si- 0.91Ge- 0.09-TFT exhibited a high-hole mobility of 112 cm 2 /V-s. In line 153,Please list the mobility of.Si1-xGex thin film. It will help other research groups to reproduce more precisely the result of your work.
- Thanks for kind suggestion. The manuscript was revised.
In this work, the field effect mobility of the poly-Si TFT was 113.81 cm2/Vs and the field effect mobility obtained by employing Ge 5.1% is 66.77 cm2/Vs.
7. In line 175, It is known that threshold voltage of p-type transistor is negative. Therefore, it should be -10.65V for the TFT with Ge 12.3%-390 mJ/cm2.
- Thanks for kind suggestion. That was a typo. The sign of the threshold voltage has been revised in the manuscript.
8.In line 204,What are the value of DOS for the case of Ge 7.1% and 10.3%?
- Thanks for kind suggestion.
Similarly, for the case of Ge 7.1%-420 mJ/cm2, the value of NTD is 6.5x1021 cm-3eV-1 and NGD is 2x1018 cm-3eV-1 and for the case of Ge 10.3%-390 mJ/cm2 , the value of NTD 8x1021 cm-3eV-1 and NGD is 4.5x1018 cm-3eV-1 .
9.In line 211, the authors state that” as the Ge concentration is increased, the grain size is decreased; and thus, the grain boundary effect related to the defect states in the energy band is increased. ” What is the distribution of DOS as a function of Ge 10.3%-390 mJ/cm2 ? Is it really higher than the case of Ge 7.1%? Please show it in Figure 5.
- Thanks for kind suggestion. We modified the distribution of DOS as a function of Ge contents and laser densities. The distribution of DOS as a function of Ge 10.3%-390mJ/cm2 is higher than the case of Ge 7.1%-420mJ/cm2.

Round 2
Reviewer 1 Report
The new sections of the paper have many examples of poor English. I have corrected some below - this is not really my job as a reviewer.
50 fabrication process.
61 ..a Ge content
62 ..the study was conducted
141 ..the real
160 As the Ge concentration increases, the..shifts to the right hand side.
195 please use the maths minus sign – not ‘dash’ – also elsewhere in the paper.
202 remove sentence ‘To improve ….poly-Si.’ It tells us nothing.
206 ‘However’ is better than ‘But’
..the Ge..
207 performance (no plural – it is a collective noun)
208 The following sentence is very poorly constructed and requires amendment.
Moreover, it was reported that the phonon band structure so that it was revealed the Si–Si peak to appear at the right position whereas the Ge–Ge 209 mode would be shifted to overestimate the Ge content.
209 ..swing is closely…
211 ..poorer than that of poly…
212 remove sentence ’The field…equation.’
215 remove. ‘.if L…are fixed.’
Can the mobility be quoted to an accuracy of two decimal places? (throughout the paper)
234 as advised last time Gaussian should have a capital letter.
243 Please use the maths multiplication symbol not ’x’
272 ‘..TFTs …were fabricated.’
In fact, the entire new paragraph requires checking by a native English speaker.
Author Response
Dear reviewer,
We appreciate you for your comments helping us to modify the manuscript (materials-470530) better for the readers of Materials.
We revised again.
Please find attached file.
Thanks.

Reviewer 2 Report
The manuscript can be accepted from my point of view.
Author Response

(The authors gave the same response as above.)
